# Red Beetroot Fermentation with Different Microbial Consortia to Develop Foods with Improved Aromatic Features

**DOI:** 10.3390/foods11193055

**Published:** 2022-10-01

**Authors:** Flavia Casciano, Hannah Mayr, Lorenzo Nissen, Andreas Putti, Federica Zoli, Andrea Gianotti, Lorenza Conterno

**Affiliations:** 1Department of Agricultural and Food Sciences (DISTAL), Alma Mater Studiorum University of Bologna, P.za Goidanich 60, 47521 Cesena, Italy; 2Laimburg Research Centre, Laimburg 6-Pfatten/Vadena, 39040 Auer/Ora (BZ), Italy; 3Interdepartmental Centre of Agri-Food Industrial Research (CIRI-AGRO), Alma Mater Studiorum University of Bologna, Via Q. Bucci 336, 47521 Cesena, Italy

**Keywords:** volatilome, *Beta vulgaris*, kefir, *Leuconostoc mesenteroides*, texture, 2-nonanone

## Abstract

The European culinary culture relies on a wide range of fermented products of plant origin, produced mostly through spontaneous fermentation. Unfortunately, this kind of fermentations is difficult to standardize. Therefore, the use of commercial starter cultures is becoming common to achieve more stable, reproducible, and predictable results. Among plant-based fermentation processes, that of the red beet (*Beta vulgaris* L. var. *conditiva*) is scarcely described in the scientific literature. In this work, we compared different types of fermentation methods of beetroot and evaluated the processes’ micro-biological, physico-chemical, structural, and volatilome features. A multi-variate analysis was used to match the production of specific VOCs to each starter and to define the correlations between the process variables and volatilome. Overall, the results showed a successful lactic acid fermentation. The analysis of the volatilome clearly discriminated the metabolic profiles of the different fermentations. Among them, the sample fermented with the mixture was the one with the most complex and diversified volatilome. Furthermore, samples did not appear softened after fermentation. Although this work had its weaknesses, such as the limited number of samples and variety, it may pave the way for the standardization of artisanal fermentation procedures of red beetroot in order to improve the quality and safety of the derived food products.

## 1. Introduction

Fermented foods are defined as “foods or beverages produced through controlled microbial growth and conversion of food components through enzymatic action [1]. There are more than 5000 different types of fermented foods worldwide [2], of which fermented meat and milk products are the most popular. On the other hand, fermented vegetables are produced in smaller amounts, and their production is often limited to a certain region and conducted on a family level [3].

There are two main methods for fermenting foods. Firstly, food can be fermented naturally, often referred to as “spontaneous ferments”, where micro-organisms are naturally present in the raw food or in the processing environment, e.g., sauerkraut, kimchi, and some fermented soy products. Secondly, foods can be fermented through the addition of starter cultures, known as “culture-dependent ferments”, e.g., kefir, kombucha, and natto [4]. Starter cultures are becoming more and more important, also in the vegetable processing sector, as they allow not only to obtain more standardized sensory and rheological properties in the final product, but also to reduce the risk of failures/disruptions and to rapidly inhibit pathogens and spoilage agents in the initial phase of fermentation [5]. Indeed, the risk of microbial growth of pathogens or spoilage agents is eliminated by the dominance of the fermentative microbial population, together with the production of organic acids and its related decrease in pH.

Lacto-fermentation is the most frequently applied type of fermentation in the transformation of vegetables. It is conducted through homo-fermentative (*Lactococcus* spp., *Pediococcus* spp., *Streptococcus* spp., and *Enterococcus* spp.) and/or hetero-fermentative (*Leuconostoc* spp., *Oenococcus* spp., and some of *Lactobacillus* spp.) lactic acid bacteria (LAB), which convert the sugars of the raw material mainly into lactic acid. Further products of their metabolism are carbon dioxide, ethanol, and acetic acid [6,7,8,9]. LAB do not only convert sugars into acids, but are also able to degrade some undesired molecules, such as toxins, phenolics, and phytates, and transform some nutrients into their more bioavailable forms, as in the case of minerals [4,10,11].

In recent years, lacto-fermented vegetables have gained more and more attraction, also as possible carriers of probiotics as alternatives to mostly dairy-based probiotic foods. Fermented products are potentially probiotic if they contain viable LAB in large numbers (6.00–8.00 Log CFU/g) [12]. To be classified as probiotics, they must withstand the adverse conditions in the gastrointestinal tract of the host (e.g., resistance to a low pH, bile acids, and enzymes), adhere and/or colonize onto intestinal cells, and prove positive health effects on the host [13].

Most scientific publications about fermented fruit and vegetable matrixes focus on large-scale products, such as kimchi, sauerkraut, table olives, and cucumber pickles. Only very few papers talk about the lactic fermentation of roots/tuberous vegetables such as red beetroot.

Red beetroots *(Beta vulgaris* L. var. *conditiva*) are widely diffused in Central and Eastern Europe, and are also cultivated in Italy. Their use is limited, and consumer acceptance is affected by their strong earthy flavour, as recently discussed by Hanson et al. [14]. Beetroots are consumed either fresh in salads or after cooking (e.g., soups) or processing (e.g., pickling and juice extraction) [15]. Beetroots contain, on average, 77.5 g/L of sugar, of which approximately 95% is sucrose and low concentrations of glucose and fructose (3 and 2%, respectively) [16]. They have an interesting nutrient profile and are rich in phenolic compounds, carotenoids, flavonoids, betalains, vitamins (vitamin A, thiamine, riboflavin, niacin, pantothenic acid, vitamin B6, vitamin C, and folate), and minerals (sodium, calcium, iron, phosphorus, potassium, magnesium, and zinc).

Most interesting is the presence of betalains, a class of natural water-soluble pigments, given their strong anti-oxidant power. Numerous bioactive compounds confer to this taproot a wide range of health-promoting properties, such as anti-cancer, anti-viral, anti-inflammatory, hepato-protective, and reduced risk of cardiovascular disease [17]. Red beetroots have been classified as a “functional food”, and their consumption is recommended to contribute to a healthy lifestyle [15,17,18].

The aim of this work is to explore the potential of red beetroot being processed via fermentation, conducting either a spontaneous or a controlled fermentation following the addition of three different types of starter cultures.

## 2. Materials and Methods

### 2.1. Vegetable Preparation

The red beetroot (*Beta vulgaris* L. subsp. *vulgaris*) cultivar Betty F1 was obtained from the experimental field of the Laimburg Research Centre, situated in the municipality of Eyrs/Oris (870 m a.s.l.) (BZ, Italy). Fresh red beetroots were immersed in tap water for approximately 10 min and then rinsed with lukewarm (30 °C) water to remove residues of soil and dirt. Vegetal parts different from roots were removed and the red beetroots were cut into small, homogeneous slices (approx. area 4 cm² × thickness 5 mm), with some disk-shaped slices (same area and thickness) used in the texture analysis. Sliced pieces were evenly distributed between 12 one-litre glass jars (Bormioli Rocco S.p.A., Parma, Italy) previously sterilized in an autoclave (120 °C for 20 min—Steam Art OT 40 L; Nüve Sanayi Malzemeleri Imalat Ve Ticaret Inc., Ankara, Turkey). A total of 650 g was added to each jar, and 500 mL of brine (NaCl 120 g/L of water) was added to obtain an overall salt concentration of 5–6% in the end-product, as recommended in previous works [19,20,21]. A plastic tool was added on top to maintain the solids submerged in the brine. Jar lids were provided with airlock valves to allow gas exhaustion.

### 2.2. Starter Cultures and Fermentation

Three different commercially available starters were compared with a spontaneous fermentation (spontaneous): a mixed culture (LAB mix) containing *Lactobacillus acidophilus*, *Levilactobacillus brevis*, *Limosilactobacillus fermentum*, *Lacticaseiobacillus paracasei*, *Lactiplantibacillus plantarum*, *Pediococcus acidilactici*, *Pediococcus pentocaseus*, and *Leuconostoc mesenteroides* subsp. *cremoris* (Sacco System by Sacco S.r.l., Cadorago, Italy); a single-strain-culture of *Leuconostoc mesenteroides* (*L. mesenteroides;* Sacco System, Italy); and a mixed culture of LAB and yeasts (Kefir; “Kefir d’acqua” Bionova S.r.l., Villanova sull’Arda, Italy). These starters were chosen amongst the ones available from the market after a preliminary screening (data not shown). The amount of inoculum established according to the manufacturer instructions was equal to 0.125 g/jar of mix or *L. mesenteroides* starter, corresponding to measurements of 9.03 and 9.21 Log CFU/mL, respectively, while 2.5 g/jar of kefir culture was added, corresponding to 8.80 Log CFU/mL of yeasts and <2.40 Log CFU/mL of LAB. Incubation was carried out at 22 °C for 19 days.

### 2.3. pH, Salt, Organic Acids, Sugars, and Alcohol Measurements

The pH was measured on days 1, 13, and at the end of fermentation (day 20) with a pH meter XS PH7 (XS-Instruments, Carpi, Italy). The end of fermentation was set at 20 days after preliminary test indicated it as the time frame necessary to reach the lowest stable pH value (data not shown). In addition, no more sugar consumption was indicated with the use of a Pocket Pal-1 refractometer (Atago Co., Ltd., Tokyo, Japan). salt concentration of the product was measured immediately after preparation and at the endpoint of fermentation with the aid of a conductivity-based pocket salt meter (Atago Co. Ltd., Tokyo, Japan). Organic acids (L-lactic acid, D-lactic acid, and acetic acid) and sugars (sucrose, glucose, and fructose) were measured using the automatic photometric analyser ItaloS (Exacta + Optech Labcenter S.p.A, San Prospero, Italy). The alcohol content was detected with a spectrophotometer (BeerLab, CDR S.r.l., Firenze, Italy). Organic acids, sugars, and alcohol were measured at the fermentation endpoint using the juice extracted from fermented vegetables (SSJ 300 A2; SilverCrest, Hoyer Handel GmbH, Hamburg, Germany).

### 2.4. Micro-Biological Analysis

A culture-dependent analysis was performed on de Man, Rogosa, and Sharpe (MRS) (Oxoid, Thermo Fisher Scientific, Waltham, MA, USA) agar medium for the LAB count, on Dichloran Rose-Bengal Chloramphenicol (DRBC) (Oxoid, Thermo Fisher Scientific, Waltham, MA, USA) agar medium for yeast count, and on Violet Red Bile Glucose Agar (VRBGA) (Oxoid, Thermo Fisher Scientific, USA) medium to determine the presence of *Enterobacteriaceae*. At the end of the fermentation, sample serial dilutions were prepared in 0.1% of pre-sterilized peptone water. Subsequently, 0.1 mL aliquots of each dilution were spread on the respective growth media for the enumeration of yeast and LAB. Petri dishes were incubated at 25 °C for 5 days in aerobiosis for yeast counts and at 30 °C for 11 days in jars with an anaerobiosis catalyst (Anaerocult A, Oxoid, Thermo Fischer Scientific, USA) for the LAB count. *Enterobacteriaceae* in the brine were determined on VRBGA medium using two agar layers. In total, 1 mL of serially diluted sample was inoculated onto the plate. Incubation was carried out at 37 °C for 24 h in micro-aerobic conditions.

### 2.5. Texture Analysis

Changes in texture properties were monitored by measuring the hardness using a texture analyser (“TA.XT plus”, Stable Micro Systems Ltd., Godalming, Surrey, UK). A penetration test was performed (cylindrical probe, flat surface, diameter 4 mm) on samples of 25 mm diameter and 5 mm thickness. The instrument was set up to run at 1 mm/s for the pre-test and test, and at 10 mm/s for the post-test periods, reaching a strain of 50 % in the samples according to the manufacturer instructions. The analysis was carried out on (a) fermented red beetroot compared with (b) raw, (c) vapour-cooked commercially available samples (Scelta Verde BioLogico, Despar, Italy), and (d) fermented and pasteurized (85 °C for 15 min.) red beetroot. Measurements were determined on: (a) and (d)—five samples for each replication jar; (b) and (c)—five samples randomly collected from several beetroot. Commercial vapour-cooked and pasteurized samples were added for the texture analysis to have a broader range and more reference of textures, since heat treatments are known to decrease hardness.

### 2.6. Consumer Test

Sensory attributes and the liking of fermented products were evaluated by a small focus-group of semi-trained panellists. The parameters (colour, odour of beetroot, earthiness, lactic acid, acetic acid, butteriness, taste of alcohol, taste, consistency/texture) were chosen before-hand and rated on a linear non-segmented scale (0–10). Fondness of the odour, taste, and overall acceptance was assessed on a 7-point hedonic scale [22].

### 2.7. Solid-Phase Micro-Extraction–Gas Chromatography–Mass Spectrometry (SPME-GC-MS)

The evaluation of VOCs was carried out on an Agilent 7890 A gas chromatograph (Agilent Technologies, Santa Clara, CA, USA) coupled to an Agilent Technologies 5975 mass spectrometer operating in the electron impact mode (ionization voltage of 70 eV) and equipped with a Chrompack CP-Wax 52 CB capillary column (50 m length, 0.32 mm ID) (Chrompack, Middelburg, NL, The Netherlands). The protocols for the solid-phase micro-extraction–gas chromatography–mass spectrometry (SPME-GC-MS) analysis and for the identification of VOCs were previously published [23,24,25,26]. Briefly, before each SPME sampling, the fibre was exposed to the GC inlet for 10 min for thermal desorption at 250 °C in a blank sample. Prior to the analysis, 6 μL of 10,000 mg/kg of 2-pentanol and 4-methyl (Merck, Darmstadt, Germany) as the internal standards were injected into the vial containing 3 mL of the liquid sample and allowed to equilibrate for 10 min at 40 °C in a water bath. The SPME fibre was exposed to each sample for 40 min and, finally, the fibre was inserted into the injection port of the GC for a 10 min sample desorption.

The temperature protocol was 50 °C for 1 min, then programmed at 1.5 °C/min to 65 °C, and, finally, at 3.5 °C/min to 220 °C, which was maintained for 25 min. Injector, interface, and ion source temperatures were 250, 250, and 230 °C, respectively. Injections were carried out in split-less mode, and helium (3 mL/min) was used as the carrier gas. Identification was obtained with the use of the NIST 11 MSMS library and the NIST MS Search program 2.0 (NIST, Gaithersburg, MD, USA). VOCs were relatively quantified from chromatogram peak areas, as a ratio peak area/total peak of different samples [27] (LOD = 0.001 mg/kg), and then sorted into chemical classes and super-normalized with the mean centring method [26]. With the samples analysed, in a volume of 3 mL, they were collected from two technical replicas of three independent experiments.

### 2.8. Statistical Analysis

For the volatilome, one-way ANOVA (*p* < 0.05) was used to determine significant VOCs among the datasets. The significant VOCs (*n* = 61) represented the total volatilome of the experiments and was reported as a quantification heatmap (Appendix A). The analyses conducted were the principal component analysis (PCA) to distribute the results on a plane and multi-variate ANOVA (MANOVA) to address specific contributions by using categorical predictors. To obtain specific correlations among molecules (VOCs) and process parameters and sensorial analysis, two independent datasets were united and computed with the Spearman rank analysis and visualized with a two-way joining heatmap including Pearson dendrograms. Statistics and graphics were created with Statistica v.8.0 (Tibco, Palo Alto, CA, USA) and the two-way joining heatmap graphic was performed with the R Expression tool on www.heatmapper.ca [28].

## 3. Results and Discussion

### 3.1. Fermentation Process Monitoring—pH, Organic Acids, Sugars, and Alcohols

The pH of the samples was measured on the day of preparation (day 1), after 13 days, and at the fermentation endpoint (day 20) (Appendix A). At the beginning (day 1), the pH of the liquid brine ranged between 6.07 and 6.30. After 13 days of fermentation, the pH decreased in all the samples and ranged between 3.55 and 3.67. The lowest final pH (day 20) was measured in the samples inoculated with kefir cultures (3.63 ± 0.01), while the highest values were measured in the *L. mesenteroides* fermentation (3.69 ± 0.03). In contrast, in the spontaneously fermented samples and in those inoculated with the LAB mix, the pH values were 3.68 ± 0.04 and 3.65 ± 0.05, respectively. At the end of the fermentation (e.g., day 20), a slight increment of pH was recorded in all samples. The pH decrease was a natural consequence of lactic fermentation, due to the production of organic acids by LAB. The reached values settled below 4.0, indicating food safety, since these pH values limited the growth of food-borne pathogens such as *Clostridium botulinum* [29]. Overall, the pH values showed a higher acidification after 13 days of fermentation and a milder one at the end. This was positive from a technological point of view, as the rapid initial acidification hindered the growth of pathogenic and spoilage bacteria, while a lower acidification at the end of the fermentation process would mean that the product could attain consumer appreciation, due to not being overly acidic [30].

The increase in pH towards the end of the fermentation and/or during the conservation of the product has been linked to the consumption of acids (lactic acid) by yeasts [31,32]. Bacteria, such as acetic acid bacteria (AAB) and some LAB, have been reported to convert Lactic acid into Acetic acid, while others, such as propionic acid bacteria, can utilize the remaining sugars or lactic acid and convert them into CO_2_ and propionic acid. All these undesired secondary fermentations lead to an increased pH [32]. The presence of *Enterobacteriaceae* in the samples was excluded by the absence of growth on the VRBG medium.

The concentration of organic acids, such as D-lactic, L-lactic, and acetic acid, as well as residual sugars was measured with the aid of enzymatic reactions, and the results are reported in Table 1. D-lactic acid was produced with the spontaneous flora 1.6 times more than with the LAB mix and *L. mesenteroides*. The amounts of L-lactic acid (LA) were generally lower than for D-lactic, and similar in all the fermented products. The overall lactic acid production was higher in spontaneous fermentation, specifically, 1.3 times higher than in the LAB mix, *L. mesenteroides*, and kefir starter fermentations Acetic acid (AA) was produced in similar amounts in the samples fermented with spontaneous flora and with kefir, with value 1.6 and 1.7 higher than in samples fermented with the LAB mix and with *Ln mesenteroides*, respectively. Even if not significantly different, the higher acetic acid production in the kefir fermentation was most likely correlated to the presence of acetic acid bacteria in this type of starter. In the spontaneous fermentation, the acetic acid concentration was similar, and we could assume that a hetero-fermentative LAB population predominated the lactic fermentation.

Compared to the recent work by Czyzowska et al. [21], the concentration of lactic acids detected in our study was lower. In fact, Czyzowska et al. [21] detected concentrations of 6.03–8.68 g/kg of lactic acid in fermented grated red beetroot, while in fermented beetroot juice, 4.76–5.18 g/L was measured after 7–10 months of storage. Further, the same study highlighted higher amounts of acetic acid, ranging from 3.54 to 6.30 g/kg and from 1.02 to 1.73 g/L in fermented grated beetroot and beetroot juice, respectively. The differences could be justified by differences in the initial sugar concentrations of the roots, the LAB species and strains carrying out the fermentation, or both.

The measurement of the residual sugars allowed for checking the status of the completion of the fermentation. Starting from an initial concentration of 58 g/L (fermentable sugars after inversion), sucrose, which normally represents approximately 95% of the total sugars in red beetroot, was exhausted (data not shown). Only residues of glucose and fructose were found in kefir (0.15 g/L), *L. mesenteroides* (0.92 g/L), the LAB mix (1.62 g/L), and spontaneous fermentation (2.44 g/L), the latter showing the highest residual together with the greatest number of organic acids (LA and AA).

The high degree of sugar consumption In the kefir samples was most likely correlated to the presence of yeasts in this starter, which, in general, showed a faster sugar metabolism than bacteria/LAB. Due to the yeast inoculum, alcoholic fermentation occurred, and the alcohol was not totally depleted by the AAB, probably due to the lack of oxygen. In fact, this sample had an alcoholic residual equal to 3.8% vol., compared to the other samples, where the alcohol content was below 1.0% vol. The higher content of LA together with the high amount of residual sugar might be linked to the contemporary presence of acids in the beetroot, such as malic acid [33], and spontaneous bacterial species capable of converting these acids into LA.

### 3.2. Micro-Biological Analysis

A culture-dependent method enumerated the LAB present in the brine at the end of fermentation after growing on MRS agar (Table 2). The highest LAB load was found in the samples fermented with the LAB mix, with values 5.7 and 5.3 times higher than those found in the samples fermented with *L. mesenteroides* and kefir, respectively.

Results clearly showed that yeast counts were high in all samples, as well as where they were not inoculated (6.06–6.82 Log CFU/mL). This led to the assumption that, somehow, the spontaneous, autochthonous microbiota of the red beetroot was able to survive and compete with the inoculated LAB strains.

In none of the samples were *Enterobacteriaceae* detected (<1.00 Log CFU/mL). As *Enterobacteriaceae* are generally considered indicators of food hygiene in European legislation, the proposed product was shown to comply with hygienic standards [34,35].

Considering the high initial inoculum (9.00 Log CFU/mL), especially for the LAB mix and the single-species LAB starter, the final LAB counts appeared to be quite low and probably having already entered the death phase. Notably, the lowest LAB load was found in samples fermented with kefir, suggesting that LAB from this starter culture were not able to tolerate the high salt concentrations, thus, classifying themselves as non-halophilic lactic acid bacteria [36]. Similar low counts (3.65–5.66 Log CFU/mL) were reported by Czyzowska et al. [21] in red beetroot fermented for two months and stored for seven months. Barath et al. [37] detected much higher LAB counts (8.00 Log CFU/mL) in beetroot juice fermented for only five days and stored for four weeks, even though the inoculum was much lower compared to the one practiced in the present work. In the cited studies, the salt concentration was much lower, 2–3% compared to 6% in the present work, and it is possible that the high salt concentration limited the proliferation of LAB in the product [38,39].

### 3.3. Texture

Besides odour and taste, texture is a very important factor that determines the overall acceptability of a food product [40]. Hardness is one of a series of mechanical properties of a material and is measurable objectively as a response of the tested material to an applied force. This was established using the penetration test, which measured the maximum force required for penetrating a tissue to a certain distance or deformation [41].

With the penetration test, net differences between the four product treatments (raw, vapour-cooked, fermented, and fermented plus pasteurized) were identified (Appendix A). As expected, the raw beetroot had the highest hardness (38.11 N), while the vapour-cooked vegetable the softest (6.72 N). In the fermented beetroot, the loss of hardness was evident, approximately 10 N lower compared to the fresh/raw product. However, the fermentation process did not cause excessive softening, being harder compared to the vapour-cooked beetroot. We measured 24.69 N, 26.18 N, 27.38 N, and 26.44 N for the spontaneously fermented beetroot, beetroot inoculated with the LAB mix, *L. mesenteroides*, and kefir, respectively. In the same fermented products, hardness was reduced to 24.42 N, 26.17 N, 23.17 N, and 26.79 N after the pasteurization treatment. Within the four different starters, no significant differences could be detected, even though a trend towards a slightly more reduced hardness was noted in the spontaneously fermented beetroot. After the pasteurization of the fermented samples, another slight decrease in hardness occurred.

The hardness of the fermented beetroot was also tested in the work by Wrzodak and Szwejda-Grzybowska [42], who obtained similar values with significant differences between the two varieties analysed. However, the hardness of the raw product was almost double (70.2–89.3 N) than that measured in our study, probably due to the varietal differences. In fact, the structural properties of fermented vegetables depend on various factors, such as the variety, degree of ripening, conservation conditions, and treatment before fermentation, salt concentration, and presence/activity of enzymes and acids. For fermented plant products, brittleness, as an index of firmness, is related to the propectin content of plants [43]. Salt seems to preserve the firmness of the tissue, probably due to being correlated to the inhibition of pectolytic enzymes or enzyme-producing micro-organisms [44]. In the presence of low sodium concentrations, propectin is hydrolysed into soluble pectic acid through the action of pectinases and cellulases, resulting in the softening of the matrix [43]. However, to avoid the negative effect of sodium intake on human health and to prevent excessive softening, Mnkeni et al. [44] suggested a short blanching treatment of approximately 15 min at 90 °C to inactivate the pectolytic enzymes (pectin-esterase and polygalacturonase), which also reduces the earthy flavour of some vegetables such as red beetroot. In our study, the softening after fermentation did not seem very high, but should be considered in the shelf-life parameters, and pasteurization can also be considered to stabilize the product.

### 3.4. Consumer Test

A consumer test was performed by a very small focus group of semi-trained panellists; therefore, only general tendencies of the chosen sensory descriptors and their acceptability could be reported in this section. The strong, earthy odour almost disappeared, whereas the typical odour of beetroot was still perceived, but at a reduced intensity compared to the fresh vegetable. This could be seen as a positive trend, making the fermented beetroot more acceptable than the raw product. The odour of lactic acid was detected in all samples and seemed to be more intense in the ones with kefir inoculum. Additionally, acetic acid was identified, although, interestingly, it was hidden somehow in the spontaneously fermented samples, where it was present at high concentrations. An alcoholic flavour was assigned to the kefir started samples, which correlated well with the results obtained via the enzymatic/spectrophotometric and SPME-GC-MS analyses. Hints of fruit and spices were perceived in the same samples, as well as in the fermentation with the LAB mix starter. Overall, the fondness of odour was higher (5.24–6.19) than for taste (3.81–4.76). The low acceptability in terms of taste was primarily influenced by the excessive saltiness of the product. No large colour changes, e.g., browning, were observed during the fermentation process, which was positive for consumer acceptance.

### 3.5. Multi-Variate Analysis of VOCs Organized by Different Chemical Classes

Through SPME-GC-MS, among four tripled cases (*n* = 12), 93 VOCs were identified with over 80% similarity to the NIST 11 MS library and the NIST MS Search 2.0 program (NIST, Gaithersburg, MD, USA). On average, 49 were found in the spontaneously fermented samples, 61 in samples fermented with the LAB mix, 58 in samples fermented with *L. mesenteroides*, and 54 in samples fermented from the kefir starter. This scenario described the entire volatilome, which was expressed by the normalized quantifications of the peak areas from each case with a heatmap of expression (Appendix A). The dataset was then sorted by chemical class (11 organic acids, 15 alcohols, 9 ketones, and 10 aldehydes), and the resulting matrices were normalized and expressed independently with the analysis of the main components (PCA). Furthermore, with the multi-variate analysis of variance (MANOVA), we assigned a significant discrimination for the type of starter culture (Appendix A).

Organic acids are some of the main fermentation products, and play a key role in the development of sensory and nutritional characteristics of the final product. A PCA of 11 statistically significant organic acids distributed the cases on the plot, separating the *L. mesenteroides* fermentation from the spontaneously, kefir-, and LAB-mix-fermented samples, which were also different from each other (Figure 1A). In our study, the main descriptors of kefir-inoculated fermentation were medium-chain fatty acids (MCFAs) such as hexanoic acid, octanoic acid, and n-decanoic acid, whereas *L. mesenteroides* was mainly characterized by nonanoic acid. Propionic acid, an important odour-active compound [45], was found mainly in kefir-fermented samples. This compound was also previously identified in a pomegranate-based drink fermented with water kefir micro-organisms [46]. It must be highlighted that fermentation with the kefir starter produced most of the esters. All these esters and higher alcohols found in the kefir-inoculated samples are associated with the yeast metabolism. Ester compounds are largely responsible for the fruity aroma associated with kefir yeast cultures [47]. Fermentation with the LAB mix was described by ethyl acetate. In fact, this molecule is mainly generated by the hetero-fermentative LAB species representing most of the species in the LAB mix starter. Ethyl acetate is used as a food flavouring agent because of its fruity aroma [48]. The results of the MANOVA discrimination for the starter cultures (*p* < 0.05) (Appendix A) indicated that fermentation with the kefir starter showed the highest percentage of most organic acids (7 out of 11). All LAB species (lactobacilli and *Leuconostoc* spp.) commonly found in water kefir, except for *Lentilactobacillus hilgardii*, were capable of producing acids from sucrose.

In particular, fermentation with the kefir starter was solely responsible for producing hexanoic acid, hexanoic ethyl ester, n-decanoic acid, decanoic ethyl ester, and propanoic acid. Fermentation with *L. mesenteroides*, on the other hand, produced almost 80% of nonanoic acid, known to impart a creamy and milky taste [49,50].

Alcohols are produced during many microbial fermentations, contributing to multiple aspects of food, such as the texture, safety, and aroma. The PCA of 15 statistically significant alcohols distributed the cases on the graph (Figure 1B). Our results showed a clear distinction between those that characterized the fermentations led by the kefir starter organisms and the spontaneous micro-organisms, while a lower difference was observed for the fermentations carried out by the LAB mix and *L. mesenteroides*. The parameters characterizing the kefir starter metabolic activity were, among others, 1-butanol and 1-pentanol, with the latter known to impart an alcoholic and fermented aroma [51]. The production of alcohols could be due to yeast activity from the kefir starter. In a recent study, it was demonstrated that soy whey fermentation conducted with water kefir generated an increase in the concentrations of esters of alcohols [52]. The fermentations with the LAB mix and *L. mesenteroides*, on the other hand, were characterized by alcohols such as 2-nonanol, 2-heptanol, and 1-octanol, while spontaneous fermentation was represented by 1-hexanol and 2-ethyl. The latter was mainly obtained through the hydrogenation of aldehydes, and had floral and fruity attributes [53]. 1-Hexanol and 2-ethyl have been found in flour or rice, soy, rye, and wheat products, and are used as flavouring agents, providing a typical pungent and green aroma [53]. The results of the MANOVA classification of the starter cultures (*p* < 0.05) (Appendix A) indicated that the fermentations guided by spontaneous micro-flora and the water kefir starter were the ones with higher alcohol concentrations than those in which fermentation was conducted only by bacteria. In particular, water kefir fermentation was solely responsible for the production of 1-butanol, 1-propanol, 1-propanol, 2-ethyl, and phenylethyl alcohol. Phenylethyl alcohol has a sweet floral taste and smell with hints of honey [54]. Fermentation with the water kefir consortium was primarily responsible for producing isoamyl alcohol (1-butanol and 3-methyl), followed by fermentation guided by the spontaneous micro-flora. Isoamyl alcohol is known for its desirable banana and pear flavours [55], and has also been identified in fennel, melon, strawberry, and tomato kefir-like beverages [56]. Isoamyl alcohol has been identified as a yeast metabolite in other fermented products [57], and this appears to have been confirmed by our results, given the increased production in the water-kefir-consortium-driven fermentation.

Ketones are produced by both bacterial fermentation and the β-oxidation of fatty acids, and their presence in fermented foods is desirable, since they give products a pleasant aroma (herbal, fruity, and floral aromas) [58]. The PCA of nine statistically significant ketones distributed cases on the graph (Figure 1C). Fermentation with the water kefir consortium was described by 2-butanone, 3-hydroxy, 2-propanone, and 1-hydroxy, while a greater speciation was observed when fermenting the beetroot with the spontaneous micro-flora, described by 2-undecanone, 2-hexanone, 2-heptanone, and 2-nonanone. As reported in a previous study [59], LAB can use citrate and pyruvate as substrates to produce 2-butanone and 3-hydroxy, which is recognized as a quality index of fermented foods through the synthesis of pyrazines. Elevated levels of 2-heptanone and 2-nonanone were found in sauerkraut samples inoculated with mixed cultures containing *Lactiplantibacillus plantarum* [60]. This result agreed with a previous finding by Liu et al. [61], who concluded that ketones were mainly produced by homo-fermentative LAB. The results of the MANOVA discrimination of the starter cultures (*p* < 0.05) (Appendix A) showed fermentation with the kefir starter to be responsible for the highest production of 2-butanone and 3-hydroxy (also known as acetoin), known to impart a sweet taste [62]. The LAB mix and spontaneous micro-flora were, instead, the only ones responsible for producing 2,3-butanedione. 2,3-Butanedione has been found in 25% of salted sauerkraut, and is formed through the decarboxylation of oxaloacetate, which originates from the metabolism of citric acid with the first step catalysed by citrate lyase [63].

Aldehydes are important compounds in fruit and vegetables, are mainly produced by the amino acid metabolism and fatty acid oxidation, contribute to characteristic fragrances and flavours, exhibit anti-microbial activity, and protect plants from pathogens [64]. Some of them are desirable because they participate positively in the aroma of the final product, such as 2-butenal, heptanal, or octanal, while others are negative due to the pungent aroma and their low threshold cytotoxicity, such as furfural and benzaldehyde [62]. The PCA of 10 statistically significant aldehydes showed cases distributed on the texture, clearly separating the fermentation with *L. mesenteroides* and that with the water kefir consortium (Figure 1D). The results showed the greatest variety of aldehydes in samples fermented with the spontaneous micro-flora, suggesting that the amino acid metabolism and lipid oxidation were more active in fermentation with indigenous micro-flora. The LAB mix and spontaneous fermentations were described by butanal, 3-methyl, and hexanal, all found in spontaneously fermented coffee beans [57]. These aldehydes have been reported as secondary metabolites of LAB and yeast as intermediates of ethanol production, and to have fruity, apple, and almond aromas [65,66]. Previous studies reported high amounts of aldehydes in yeast-inoculated fermented foods [66,67]. In any case, the presence of these volatile substances could contribute to the high aromatic score and fruity aroma of the fermented product. The main descriptors of kefir fermentation were benzaldehyde, 2-methyl, and benzaldehyde. Some volatile compounds, including benzaldehyde, were present only due to some specific LAB. In fact, benzaldehyde production has been previously reported only for *Lactobacillus helveticus* [62], while the production of 2-heptanal, which gives an acidic aroma, is produced by *Lb. plantarum* and *Campanilactobacillus farciminis* [62]. The results of the MANOVA classification of the starter cultures (*p* < 0.05) (Appendix A) indicated that the greatest production of aldehydes was measured after the fermentation guided by the spontaneous micro-flora and the LAB mix, while the production of this class of compounds was reduced in beetroot fermented after the addition of the kefir starter.

### 3.6. Spearman Rank Correlations

Spearman rank correlations (*p* < 0.05), two-way joining heat maps, and the Pearson cluster analysis were performed by comparing two different normalized datasets, each derived from relative quantification values of the entire experimental dataset (Figure 2). The significance of the correlations is shown in Appendix A. Two main clusters were identified from Pearson’s left dendrograms. The two clusters were then distributed into two different subgroups defined by the top-side dendrograms. Cluster one hosted correlations within specific VOCs and three process variables, namely, the acidification trend, the NaCl content, and the hardness intensity. Subgroup 1A hosted positive correlations, while 1B hosted negative ones. In subgroup 1A, the most significant and interesting correlations were those related to ten different VOCs in respect to acidification values. Amongst these ten VOCs, three were organic acids and three were derivative esters, which could have been the ones to mostly contribute to the reduction in pH. The most important volatile organic acids were hexanoic and octanoic acids. In this cluster, it was also interesting to note that acetoin was positively correlated to hardness. In fact, this compound in baking is also used to improve the structure and texture of foods [62]. In particular, the PCA of the volatilome highlighted the octanoic and hexanoic acids and acetoin as specific descriptors of beetroot fermented with the kefir starter. Kefir had the top acidification trend and the second-best hardness score. Hexanoic and octanoic acids have many positive effects, such as being anti-microbial and anti-oxidant [68], thus, able to improve the overall food safety and extend the food’s shelf-life [62], and being beneficial for humans [69], are known for their undesirable aroma and low-odour threshold [62]. This consideration could also help to understand the negative correlation found among purchasability and this group of volatile organic acids. Indeed, the beetroot fermented with kefir was identified by the lowest appreciation, odour, and taste scores. Cluster 2A hosted negative correlations that strengthened our experimental approach and the dataset results, as fermentation alcohols, such as 1-pentanol, 1-hexanol, or 2-heptanol, were inversely correlated to acidification. The sensorial features related to odour and taste satisfaction were, instead, better defined in subgroup 2B by a set of VOCs with significant positive correlations, namely, 2-undecanone, 2-nonanone, and 2-heptanol. The PCA pointed out how these VOCs specifically described the fermentation guided by the spontaneous micro-flora, and the fermented product achieved the best score for the odour parameter. These VOCs were characterized by an aroma defined as being fruity, creamy, and herbaceous [70].

## 4. Conclusions

The basic parameters measured from the fermented beetroot highlighted successful lactic fermentations, even in the non-inoculated samples. Even if the supremacy of the added starters was not established and some metabolic action from the spontaneous micro-flora occurred, the volatilome showed a clear distinction of the metabolic profiles of different fermentations. In particular, *L. mesenteroides* had a distinct one described by the lowest diversity of molecules. The samples fermented with the kefir starter were characterized by metabolites associated mainly with yeast fermentation, such as isoamyl alcohol and phenylethyl alcohol. The samples fermented with the LAB mix showed a higher complexity of the volatilome, as the one fermented with a mixture of micro-organisms. The fermentation guided by the spontaneous micro-flora was mostly described by molecules with positive aromatic notes, such as 2-undecanone and 2-nonanone. The texture analysis showed that fermented samples were not excessively softened, and should be considered for a shelf-life evaluation. From this point of view, pasteurization can be suggested.

Raw fruit and vegetables are foods of high nutritional and functional value. However, due to their short shelf-life, a large amount of them is discarded as waste, generating large economic losses and the accumulation of organic waste. The production of fermented plant-based foods and beverages, therefore, represents a viable alternative for their sustainable use. Even if further studies should be carried out to better establish a successful process and product shelf-life, and the possibility to use a lower amount of salt, this study highlighted that this method could be a promising way to preserve beetroot and its nutritional value.

## Figures and Tables

**Figure 1 foods-11-03055-f001:**
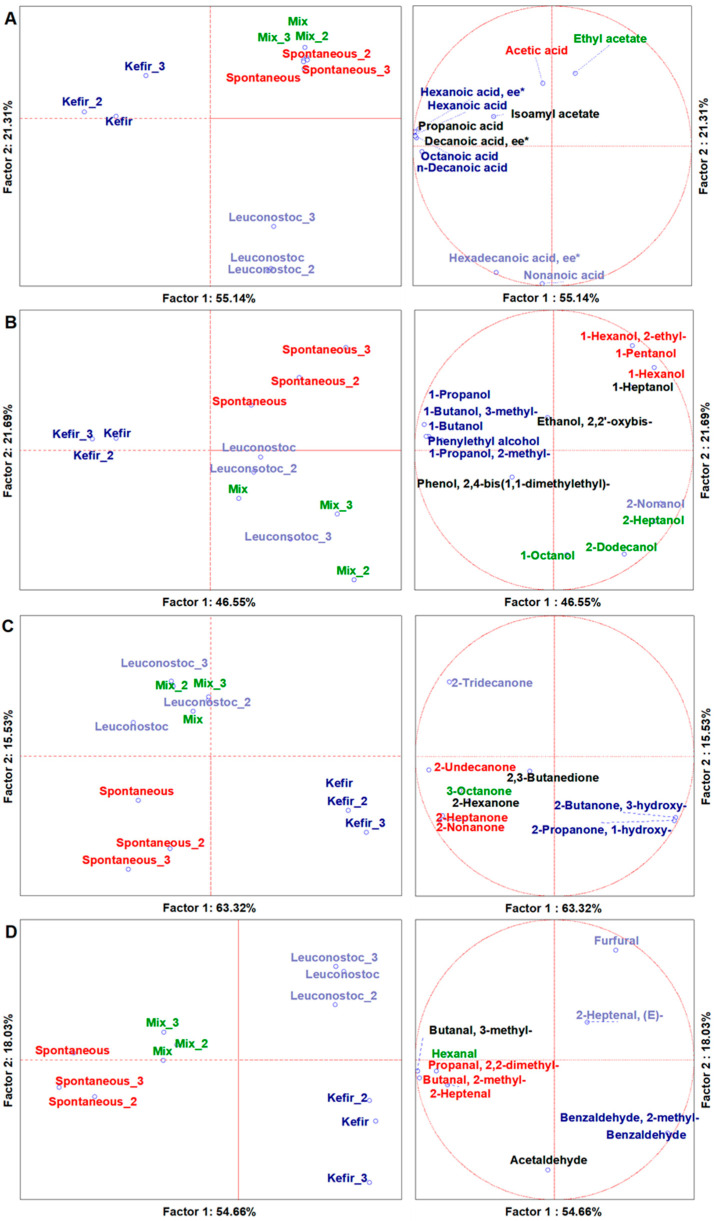
PCAs of the volatilome sorted by chemical classes of significative (ANOVA *p* < 0.05) VOCs. (**A**)—organic acids; (**B**)—alcohols; (**C**)—ketones; (**D**)—aldehydes. Left-side diagrams are for PCAs of cases, while right-side diagrams are for PCAs of variables. * ee—ethyl ester; _2 and _3 indicate the mean of technical replicas of experimental replicates. Mix—sample fermented after LAB mix starter addition; Spontaneous—sample fermented with spontaneous micro-organisms; *Leuconostoc*—sample fermented after *L. mesenteroides* starter addition; Kefir—sample fermented after water kefir starter addition.

**Figure 2 foods-11-03055-f002:**
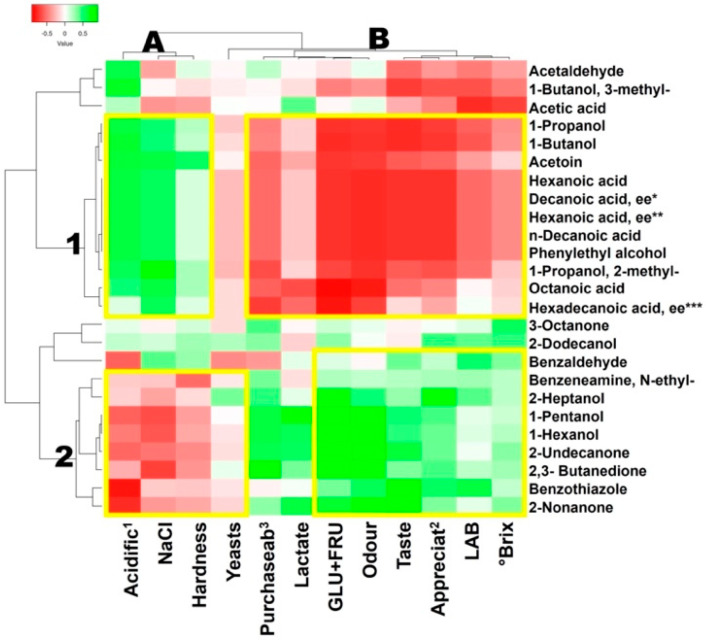
Spearman rank correlations among process parameters and VOCs. Dendrograms were determined with Pearson analysis with complete linkage. * Significance of correlations determined with Spearman rank analysis is reported in Appendix A (*p* < 0.05). A,B. top clusters; 1,2: side clusters. Acidific^1^—acidification; Appreciat^2^—appreciation; Purchaseab^3^—purchasability. NaCl indicates the salt content; LAB—lactic acid bacteria; GLU + FRU indicates absolute quantification of glucose and fructose; lactate indicates absolute quantification of lactic acid; °Brix indicates the sugar content. * Decanoic acid, ethyl ester; ** hexanoic acid ethyl ester; *** hexadecanoic acid ethyl ester.

**Table 1 foods-11-03055-t001:** Mean concentrations (g/L ± standard deviation) of organic acids and residual sugars at the endpoint of fermentation.

	D-Lactic Acid	L-Lactic Acid	Lactic Acid Tot.	Acetic Acid	Res. Sugars
**Spontaneous**	2.94 ± 0.35 ^cA^	1.63 ± 0.24 ^bcA^	4.57 ± 0.46 ^aA^	0.53 ± 0.17 ^bA^	2.44 ± 0.96 ^cA^
**LAB mix**	1.76 ± 0.38 ^bB^	1.56 ± 0.10 ^bA^	3.32 ± 0.47 ^aB^	0.32 ± 0.04 ^cA^	1.62 ± 0.61 ^bAB^
** *L. mesenter.* **	1.91 ± 0.39 ^bB^	1.39 ± 0.18 ^bA^	3.31 ± 0.57 ^aB^	0.30 ± 0.02 ^bA^	0.97 ± 0.52 ^bAB^
**Kefir**	2.12 ± 0.08 ^bAB^	1.24 ± 0.09 ^cA^	3.36 ± 0.16 ^aB^	0.54 ± 0.07 ^dA^	0.15 ± 0.12 ^eB^

Different letters indicate statistical significance ^abcd^ within the same treatment or ^AB^ between treatments after ANOVA followed by Tukey’s HSD (honestly significant difference) post hoc test (*p* < 0.05). *L. mesenter.*—*Leuconostoc mesenteroides*; Lactic acid tot—sum D- and L-lactic acid total; Res. Sugars—residual sugars.

**Table 2 foods-11-03055-t002:** Mean counts (Log CFU/mL ± standard deviation) of yeast and LAB (lactic acid bacteria) enumerated by plate count method.

	LAB (Log CFU/mL)	Yeasts (Log CFU/mL)
**Spontaneous**	5.18 ± 4.78 ^b^	6.23 ± 6.39
**LAB Mix**	6.07 ± 4.76 ^c^	6.82 ± 6.81
** *L. mesenteroides* **	6.24 ± 5.91 ^a^	6.06 ± 5.96
**Kefir**	5.07 ± 5.06 ^b^	6.09 ± 6.02

^a,b,c^ Different letters indicate statistical significance determined with ANOVA, followed by Tukey’s HSD (honestly significant difference) post hoc test (*p* < 0.05).

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
