# Peer review of "Red Beetroot Fermentation with Different Microbial Consortia to Develop Foods with Improved Aromatic Features"

_foods, 2022, doi:10.3390/foods11193055_

Round 1

Reviewer 1 Report

The experimental procedure carried out in the fermentation of red beet is adequately guided, the data obtained are well discussed and the presentation in the heat maps and in the principal component analysis allows an adequate understanding of the information.

Author Response

Response to reviewer comments

We thank all the reviewers for their thoughtful/useful comments and suggestions.

Reviewer 2 Report

The study by Casciano et al. concerns the characterization of the fermentation of red beetroots.

The study is straightforward. 

Please find below some comments for the authors' consideration:

1. Line 13:  Instead of “European food culture” it could be better to state “European culinary culture” since you mention homemade fermentations.

2. Line 14: “…by spontaneous fermentation, which results difficult to standardize.” This sentence is not correct. Please revise.

3. Line 19: “to match the production of specific VOCs to each starter and to define correlations between process variables and the volatilome.”

4. Line 21: “lactic acid fermentation”

5. Line 21: “The analysis of the volatilome clearly distinguished the metabolic profiles of the different fermentations, by the higher complexity of the volatilome” Complicated. Please revise.

6. Line 146: “were spread on the respective growth media

7. Line 147: “Petri dishes” ?

8. Line 163: “c) vapour-cooked commercially available samples (Scelta Verde BioLogico, Despar, Italy) and d) fermented and pasteurized (85 °C for 15 min.)” It is not clear why did the authors include all these samples in the comparison test. Raw vs fermented would have been ok.

9. Paragraph 3.3 needs to be shortened. Please delete excessive background information.

10. The same applies for Paragraph 3.5

11. Paragraphs 3.5. and 3.6 need to be joined in a single paragraph.

12. Please revise the whole document to correct minor syntax and grammar errors. Please make sure that each sentence you include in the final text communicate the message you want correctly.

Author Response

Response to reviewer comments

We thank all the reviewers for their thoughtful/useful comments and suggestions.

Their comments have improved the manuscript effectively. We have included all of their suggestions and below we present a point-by-point response to their comments (Q).

A detailed list (point to point) of responses (A) to each item of the review comments.

Changes in the manuscript Foods-1865532 R1 track have been highlighted in red

To Reviewer 2

Q(x) Moderate English changes required

R-English grammar, fluency, and typos were corrected

Q1. Line 13:  Instead of “European food culture” it could be better to state “European culinary culture” since you mention homemade fermentations.

A1.We thank the reviewer for the comment, and we have changed the term accordingly.

Q2. Line 14: “…by spontaneous fermentation, which results difficult to standardize.” This sentence is not correct. Please revise.

A2.The sentence has been modified and splitted into two sentences

Q3. Line 19: “to match the production of specific VOCs to each starter and to define correlations between process variables and the volatilome.”

A3.We have revised the sentence.

Q4. Line 21: “lactic acid fermentation”

A4.We have modified as suggested.

Q5. Line 21: “The analysis of the volatilome clearly distinguished the metabolic profiles of the different fermentations, … by the higher complexity of the volatilome” Complicated. Please revise.

A5.We have modified the sentence, to clarify the meaning for a better understanding.

Q6. Line 146: “were spread on the respective growth media”

A6.We modified the sentence accordingly, deleting “by spatulation

Q7. Line 147: “Petri dishes” ?

A7.We modified as suggested

  1. Line 163: “c) vapour-cooked commercially available samples (Scelta Verde BioLogico, Despar, Italy) and d) fermented and pasteurized (85 °C for 15 min.)” It is not clear why did the authors include all these samples in the comparison test. Raw vs fermented would have been ok.

A Line 163-We thank you the reviewer for the comments. We chose these two additional products for the texture analysis to have a broader range and more reference for texture, since heat treatment are known to decrease hardness. We added this explanation in the text for a better understanding “Line 171-173”.

Q9. Paragraph 3.3 needs to be shortened. Please delete excessive background information.

A9.We thank the reviewer for this comment. The excessive background was deleted and only information pertinent to the discussion were left.

Q10. The same applies for Paragraph 3.5

A10.We thank the reviewer for this comment. The paragraph has been shortened as suggested.

Q11. Paragraphs 3.5. and 3.6 need to be joined in a single paragraph.

A11.We thank the Reviewer for this comment, but we would like to maintain the structure of these sections as separated, because we consider these two aspects of high relevance.

Q12. Please revise the whole document to correct minor syntax and grammar errors. Please make sure that each sentence you include in the final text communicate the message you want correctly.

A12.We thank you the reviewer for its comment. The whole document has been checked and revised.

Reviewer 3 Report

Dear editor,

I have revised the article "Red Beetroots fermentation with different microbial consortia to develop foods with improved aromatic features" and some issues must be adressed:

- Correct the citation format throughout the paper: for example (Ojha e Ti- 31

wari, 2016) [1].

-Update some references, especially in the introduction section, because some of them are more than 10 years old.

-Please rewrite the introduction section: the information was written in a single paragraph and in a non-logical sequence.

-section 2.2: Why did you use a mixture of LAB and not use each strain separately?

-line 150: rewrite the overlay trial used to determine Enterobacteriaceae. Was it MRS + VRBGA or PCA + VRBGA? 1 mL of serial-diluted brine?

-Line 238: Could not the higher pH at 20 days mean the growth of a spoilage organism? Why did you decide to end the experiment on day 20, since fermented vegetables can last more than 2 to 3 months?

258-261: I have noticed that the production of some organic acids is higher in some samples. So I wonder whether or not there is statistical difference. When we look at Table 1, we can realize that the authors compared metabolite production and did not compare the same metabolite content between the different fermentation processes. Therefore, making a comparison between treatments would be more sensible in this case.

-line 262: Compared to the recent work of Czyzowska et al. (2020) [23],

-Table 2: Convert the results in Log CFU/ mL.

Author Response

Response to reviewer comments

We thank all the reviewers for their thoughtful/useful comments and suggestions.

Their comments have improved the manuscript effectively. We have included all of their suggestions and below we present a point-by-point response to their comments (Q).

A detailed list (point to point) of responses (A) to each item of the review comments.

Changes in the manuscript Foods-1865532 R1 track have been highlighted in red

To Reviewer 3

Q(x) English language and style are fine/minor spell check required

A English grammar, fluency, and typos were corrected

Q- Correct the citation format throughout the paper: for example (Ojha e Ti- 31

wari, 2016) [1].

A-We thank you the reviewer this comment. We delete the non-required format.

Q-Update some references, especially in the introduction section, because some of them are more than 10 years old.

A-We thank you the reviewer for the comment. In the revised version we have updated the bibliography adding more recent references and deleting those “old” when not essential.

Q-Please rewrite the introduction section: the information was written in a single paragraph and in a non-logical sequence.

A-We thank the reviewer and in the revised version a better structure was given to the introduction, as suggested.

Q-section 2.2: Why did you use a mixture of LAB and not use each strain separately?

A-section 2.2 We thank you the reviewer for the comment. The intent of the study was to compare commercially available starters, therefore we took some of the one available from the market as they were produced. We tried to better clarify this concept in the revised version of the manuscript (Line 110,17-118).

Q-line 150: rewrite the overlay trial used to determine Enterobacteriaceae. Was it MRS + VRBGA or PCA + VRBGA? 1 mL of serial-diluted brine?

A-We thank you the reviewer for this comment. Enterobacteriaceae were determined on VRBGA medium using the double layer technique. 1 mL of serially diluted brine sample at the end of the fermentation was inoculated into the plate. We modify the paragraph for a better understanding (Line 149,154-155).

Q-Line 238: Could not the higher pH at 20 days mean the growth of a spoilage organism? Why did you decide to end the experiment on day 20, since fermented vegetables can last more than 2 to 3 months?

A-We thank you the reviewer for this comment. Since not a lot of literature about beet root fermentation has been found we ran a preliminary experiment to set the fermentation end as the decrease of pH to a stable value. We also tested the sugar consumption measuring the Brix (data not shown). The aim of our work was to establish the possibility to transform beetroot by lacto-fermentation, and the effect of using commercially available starters. The starter effectiveness assessed the possibility to reach the acidification level necessary for food-born pathogen exclusion. We did not plan a shelf-life test for the product, also due to the length and cost of this kind of test. We only planned to exclude the presence Enterobacteriacee as possible cause for acid consumption and related pH increase. We added this comments in the revised manuscript for a better understanding (Line 128-131).

Q-258-261: I have noticed that the production of some organic acids is higher in some samples. So I wonder whether or not there is statistical difference. When we look at Table 1, we can realize that the authors compared metabolite production and did not compare the same metabolite content between the different fermentation processes. Therefore, making a comparison between treatments would be more sensible in this case.

A-We thank you the reviewer for this comment. The comparison between treatment has been added

Q-line 262: Compared to the recent work of Czyzowska et al. (2020) [23],

A-We thank you the reviewer for this comment. A comparison to the recent work suggested has been added to the manuscript (line 270-277).

Q-Table 2: Convert the results in Log CFU/ mL.

A-Table 2 and the data values contained in the text of were revised accordingly

Round 2

Reviewer 3 Report

Dear editor,

The authors have addressed all the issues raised, so that the MS is now fit for publication.

Best regards